# Ion Channels as Therapeutic Targets in High Grade Gliomas

**DOI:** 10.3390/cancers12103068

**Published:** 2020-10-21

**Authors:** Michaela Griffin, Raheela Khan, Surajit Basu, Stuart Smith

**Affiliations:** 1Children’s Brain Tumour Research Centre, Biodiscovery Institute, University of Nottingham, Nottingham NG7 2RD, UK; michaela.griffin@nottingham.ac.uk; 2Division of Medical Sciences and Graduate Entry Medicine, Royal Derby Hospital, University of Nottingham, Nottingham NG7 2RD, UK; Raheela.khan@nottingham.ac.uk; 3Department of Neurosurgery, Queen’s Medical Centre, Nottingham University Hospitals, Nottingham NG7 2RD, UK; Surajit.Basu@nuh.nhs.uk

**Keywords:** ion channel, glioblastoma multiforme, ion channel inhibitor, membrane potential, glioma

## Abstract

**Simple Summary:**

Glioblastoma multiforme is an aggressive grade IV lethal brain tumour with a median survival of 14 months. Despite surgery to remove the tumour, and subsequent concurrent chemotherapy and radiotherapy, there is little in terms of effective treatment options. Because of this, exploring new treatment avenues is vital. Brain tumours are intrinsically electrically active; expressing unique patterns of ion channels, and this is a characteristic we can exploit. Ion channels are specialised proteins in the cell’s membrane that allow for the passage of positive and negatively charged ions in and out of the cell, controlling membrane potential. Membrane potential is a crucial biophysical signal in normal and cancerous cells. Research has identified that specific classes of ion channels not only move the cell through its cell cycle, thus encouraging growth and proliferation, but may also be essential in the development of brain tumours. Inhibition of sodium, potassium, calcium, and chloride channels has been shown to reduce the capacity of glioblastoma cells to grow and invade. Therefore, we propose that targeting ion channels and repurposing commercially available ion channel inhibitors may hold the key to new therapeutic avenues in high grade gliomas.

**Abstract:**

Glioblastoma multiforme (GBM) is a lethal brain cancer with an average survival of 14–15 months even with exhaustive treatment. High grade gliomas (HGG) represent the leading cause of CNS cancer-related death in children and adults due to the aggressive nature of the tumour and limited treatment options. The scarcity of treatment available for GBM has opened the field to new modalities such as electrotherapy. Previous studies have identified the clinical benefit of electrotherapy in combination with chemotherapeutics, however the mechanistic action is unclear. Increasing evidence indicates that not only are ion channels key in regulating electrical signaling and membrane potential of excitable cells, they perform a crucial role in the development and neoplastic progression of brain tumours. Unlike other tissue types, neural tissue is intrinsically electrically active and reliant on ion channels and their function. Ion channels are essential in cell cycle control, invasion and migration of cancer cells and therefore present as valuable therapeutic targets. This review aims to discuss the role that ion channels hold in gliomagenesis and whether we can target and exploit these channels to provide new therapeutic targets and whether ion channels hold the mechanistic key to the newfound success of electrotherapies.

## 1. Glioma

Gliomas are tumours that arise from glial precursor cells originating from the brain and the spinal cord. These glial neoplasms comprise a sizeable group of tumours that can be classified into histological, molecular and clinicopathologic subtypes [1]. Gliomas are classified as low grade (WHO grade I/II) and high grade (WHO grade III/IV), with glioblastoma (multiforme) (GBM) being an aggressive malignant WHO grade IV astrocytoma. The WHO 2016 classification was adapted to provide more comprehensive molecular subgrouping of gliomas and now includes 1p/19q-codeletion (oligodendroglioma), isocitrate dehydrogenase (IDH) mutations and H3K27M mutants [2]. It is now thoroughly recognised that gliomas are a not a single entity, but a heterogeneous group of tumours associated with very well-established subtypes that alter in outcome and incidence relative to age. GBM has been classified on the basis of gene expression as four distinct subgroups: proneural, neural, classical and mesenchymal [3]. Further delineation can be provided by genome wide approaches such as utilising DNA methylome arrays [4,5].

GBM has a global incidence of 10 per 100,000 of the population and can affect people of all ages, although peak age of diagnosis falls between 45 and 75 years [6]. Primary GBM (those that arise de novo) account for 95% of tumours, whereas those arising from precursor less malignant gliomas (secondary, usually with an IDH mutation) account for the remaining 5% [7]. Treatment prospects are bleak for GBM; initial surgical intervention is the main predictor of outcome and is necessary to gain a clear histological diagnosis for the glioma. Despite this, complete resection is rarely accomplished due to the aggressive and invasive nature of GBM cells. Infiltrative disease remains within adjacent brain tissue and is responsible for tumour regrowth [8]. Concomitant alkylating chemotherapy (temozolomide) and ionizing radiation follows surgery but often has limited effect on GBM progression [3].

## 2. Ion Channels

The transports of ions across the cell membrane is a fundamental process in maintaining normal cellular function and activity. Ion channels contribute to the cell cycle, cell death [9], cell volume regulation and intrinsic proliferative capacity; all of which are vital to cell survival [10]. The transport of ions across the membrane is critical in both normal and tumour cell survival and may be a factor in progression from normal to malignant state [11].

Mounting exploratory evidence suggests that ion channels not only regulate the electrical signaling of excitable cells, but they also play a crucial role in the progression of brain tumours [12]. It’s becoming apparent that cancers of the nervous system cross talk, systematically and within the local tumour microenvironment. Communication (via synapses) between cancer cells and neurones utilises neurotransmitters and voltage gated mechanisms to regulate cancer cell growth [12]. Further to this, glioma cells can electrically integrate into neural circuits through neurone-glioma synapses [13]. Ion channels function in a plethora of regulatory pathways, including those important in tumour vascularisation, and tumour-immune cell interactions [14]. Ion channel dysregulations are not a recognised cancer hallmark, however their activity is likely to underlie several known hallmarks such as proliferative capacity, apoptotic avoidance and invasion.

### 2.1. Ion Channels and Membrane Potential (V_m_)

Ion channels and transporters have a primary role in generating cellular membrane potential (V_m_). Ion channels function as selectively permeable pores, allowing ions to cross the membrane according to chemical and electrical gradients [15]. The V_m_ of a resting cell is negative; when the membrane potential is moved to a more negative state, the cell is hyperpolarised, and when the V_m_ moves to a less negative state, the cell is depolarised [15]. Membrane potential arises due to a difference in electrical charge on either side of a cell membrane; a direct result of ion diffusion and electrogenic pumps. In animal cells it is the passive movement of Na^+^, K^+^, Ca^2+^ or Cl^−^ ions based on their corresponding electrochemical gradients that contribute most to the electrical potential of the membrane [16].

Voltage gated ion channels (VGICs) form a distinct group of channels that react to changes in membrane potential. VGICs are selectively permeable to Na^+^, K^+^, Ca^2+^ and Cl^−^ ions. In excitable cells, VGICs function to generate action potentials in neurons and contractions in muscle [15]. They also play a pivotal role in controlling ion movement, maintaining homeostasis and thus regulating proliferation and cell volume [17] Figure 1 depicts the movements of ions through VGICs throughout the duration of the cell cycle.

The intricacies of ion channel regulation are outside the scope of the review, however there are four noted levels of ion channel regulation: gene transcription, trafficking and subcellular localisation of the channel proteins, alternative splicing and post translational modification [18]. Whilst much research is still undergoing to determine the transcriptional regulation of ion channels, studies have found that the expression of SCN3A is regulated by methylation of promoter CpGs [19]. Similarly, methyl-CpG-binding domain protein 2 (MBD2) has been implicated in VGSC regulation; MBD2 targets CPGs, which may lead to transcription. Recently, RACK1 was found to suppress SCN1A expression [20]. Similarly, Repressor element 1-silencing transcription factor (REST or NRSF) effects the expression of voltage gated sodium channels (VGSCs), repressing Nav1.2 gene. A c-terminal fragment of the Cav1.2 channel (calcium channel-associated transcription regulator CCAT) translocates to the nucleus in neurones, regulating gene transcription through interactions with p54 and connexin 31.1 [18]. Alternative splicing, particularly in voltage gated channels provides ion channels with distinct kinetics. One of the major molecular mechanisms that influence VGCs is protein phosphorylation, particularly by PKA and PKC [18].

### 2.2. Ion Channels in Cell Cycle Progression

It is well established that the control of the cell cycle accounts for the proliferative capacity of a cell [21] and increasing data harnessed from bioelectric studies demonstrates an important role for membrane potential in cell cycle activity [22,23]. Stringent regulatory measures are employed to maintain cellular homeostasis, whereby a multistep and rhythmic pattern of hyperpolarisation and depolarisation of cellular V_m_ drives cells through their cycle [24]. Holding the key to coordinated cell interactions, this unique and powerful signalling system is well conserved, but poorly understood as a driving force of cancers.

Cancer genotypes can be described by common features; defined in 2000 and revisited in 2011 [25,26] as the ‘hallmarks of cancer’. These developing clinicopathologic features are a backbone of cancer research, setting precedent for all areas of study. Amongst the genes linked to cancer, those encoding ion channels stand out. The dysregulation of the homeostatic maintenance of intracellular ionic function underpins many of the pathophysiological events that define these hallmarks [27], thus there is strong motive to consider ion channel expression as a signature of cancer progression.

The part that ion channels play in cancer was first established in a series of seminal experiments by Cone and colleagues. They observed that sarcoma cells underwent hyperpolarisation before entering M phase, indicating that membrane potential may play a key role in cell cycle progression [28]. To consolidate this evidence, a later study [29] revealed that hyperpolarisation blocked mitosis and subsequent DNA synthesis in a reversible manner. From these data it was postulated that the V_m_ of a cell was correlated to its state of differentiation, for example, terminally differentiated cells such as epithelial cells possessed a hyperpolarised V_m_ [30]. These experiments provided a platform for further investigation into a now well understood phenotype: highly plastic cells such as tumour cells and embryonic cells retain a depolarised state [31], whereas quiescent cells tend to be hyperpolarized [32].

Studies quickly established that there is significant depolarisation of the V_m_ during malignant transformation of normal cells. Akin to Cone’s theory of cellular V_m_ [30] many in vivo and in vitro studies e.g., those of normal breast and breast cancer cells [33] normal hepatocytes versus hepatocellular carcinoma [34], demonstrated that cancer cells tend to be more depolarised than their normal counterparts [24]. As previously noted, ionic exchange processes are responsible for proliferative activity, apoptosis, and migration of cells. The link between ion channels and apoptosis is extensive, one such clear link is demonstrated by the role of Ca^2+^ channels. Calcium channels are implicated in apoptotic pathways. Cytoplasmic Ca^2+^ overload triggers apoptosis by differing pathways. Increased Ca^2+^ levels promote mitochondrial uptake of Ca^2+,^ opening MPTP and triggering the intrinsic pathway. Calpains are Ca^2+^-dependant cysteine proteases that mediate BCL-2 family cleavage, including BID and BCL-2 and similarly promote the release of both Cyt C and MOMP [35].

The role that ion channels play in carcinogenesis was first understood when small cell lung cancer cell lines were observed to exhibit unusual patterns of ion channel functions [36] and that, when subjected to pharmacological intervention, cancer cell growth was inhibited [37]. This provided evidence that disordered function or expression of ion channel genes contributed widely to neoplastic progression of cells, triggering a new milieu of research including clinical trials on targeting ion channels in basal cell carcinoma [38]. A summary of ion channels as antibody targets is summarised by Hutchings and colleagues [39]. 

## 3. Ion Channels in Glioma

### 3.1. Ion Channels in Invasion and Metastasis of Glioma Cells

Glioma cells appear to have specific invasive capabilities and adapt to spatial constraints; an elongated spindle-like morphology aids in the invasion of glioma cells into the surrounding brain parenchyma [40]. This morphological change relies on cellular shrinkage, a mechanism regulated by Cl^−^ and K^+^ mediated efflux of water [41]. During apoptosis, a cell undergoes cell shrinkage, a hallmark referred to as apoptotic volume decrease (AVD). Altered regulation of cell volume and morphology is associated with apoptosis in both normal and malignant cells. Studies have shown that a down-regulation of these channels is associated with the ability of cancer cells to evade apoptosis [42]. An emerging model for the remarkable way that glioma cells undergo volume changes, coined the hydrodynamic model, focuses on a coordinated cascade, initiated by the ligand-induced activation of ion channels. The resulting movement of water along with Cl^−^ and K^+^ ions facilitates morphological alterations, allowing glioma cells to successfully navigate the constricted environment of the brain [43]. The specific roles that ion channels play in glioma is summarized in Table 1.

Anions are particularly crucial in the migratory capacity of GBM. In glioma cells, the cotransporter activity of Na^+^, K^+^ and 2Cl^−^ leads to the accumulation of intracellular Cl^−^ (up to 100 mM) [44], the sizable conductance of Cl^−^ and its activity lead to membrane depolarisation. This depolarisation follows the exit of the anions, driving K^+^ exit from the cell. Cumulative cellular loss of KCl and various osmolytes results in cellular shrinkage [11]. A characteristic loss of cell volume precedes M phase and is known as ‘pre-mitotic condensation’ [45]. Chloride channel blockade prevent the loss of Cl^−^ from the cell, and therefore the morphological changes associated with premitiotic condensation [46].


cancers-12-03068-t001_Table 1Table 1Ion channels in glioma.Channel TypeCell/Tumour ModelEffect on GliomagenesisReferencesEag1 (Kv10.1)GBM Cell LinesHuman GliomaSuppression of Eag1 sensitises GBM cells to TMZ. Gliomas, despite of their grade, tend to overexpress Eag1Kv10.1 expression confers a significantly longer overall survival [47,48,49]NaV1.6Nav1.1Human GliomaGBM Cell LineHigh expression in glioma tissue compared to normal brain. Knock down of SCN8A decreases glioma cell viability. NaV1.1 and NaV1.6 play role in cytokines release in glial cells[50,51]TRPM3GBM cell linesHigh expression of TRPM3 linked to decreased median survival[50,52]P2RX4Human gliomaHigh expression of P2RX4 linked to decreased median survivalSilencing suppresses glioma cell growth through BDNF/TrkB/ATF4 signaling pathway[50,53]CLCN3GBM cell linesReduced expression of CLCN3 inhibits migration of GBM cellsCLCN3 suppression can sensitize glioma cells to cisplatin through lysosomal dysfunction[54,55]CLCN6Human GliomaDown regulated in human glioma, significantly increased risk of death.[54]CLIC1Human GliomaGBM cell linesGlioma stem cellCLICL1 is up-regulated in human glioma, conferred poor overall survivalCLIC1 silencing reduced proliferative, clonogenic, and tumorigenic capacity of stem/progenitor cellsInhibition of CLIC1 at G1/S transition by metformin is a has an antiproliferative effect in glioblastomaBiguanide inhibition impairs GSC viability, invasiveness, and self-renewal[54,56,57,58]CLIC4Human GliomaGBM cell linesDown regulated in human glioma, significantly increased risk of death.Knockdown of CLIC4 enhances ATP-induced HN4 cell apoptosis through mitochondrial and endoplasmic reticulum pathways[54,59]P2RX7Human GliomaDown regulated in human glioma, significantly increased risk of death.P2 × 7 receptor antagonism inhibits tumour growth [54,60,61]VDAC2Human GliomaHighly expressed in glioma tissues[54]SLC12A1GBM cell linesOverexpression inhibits glioma cell proliferation[62]ENaC GBM cell linesEnhances glioma motility.Toxin inhibits whole cell current in GBM cellsγENaC subunits present in glioma samples, but not healthy astrocytes.[62,63]ASIC1 GBM cell linesInvolved in glioma cell shrinkage, enhancing invasive capacity.Psalmotoxin inhibits whole cell currents in GBM cellsMambaglin-2 inhibits cell growth[63,64,65,66,67]TRPC6GBM cell linesMediator of notch driven invasiveness in gliomaKnock down of gene inhibits invasion[68]AQP1GBM cell linesHigh expression enhances migration[69]Kir1.4GBM Cell linesOverexpression halts glioma cell division[70]ClC-2, -3 & -5GBM Cell LinesHuman GliomaHigh expression levelsMediates cell shrinkage of invading cellsCIC3 is a critical regulator of the cell cycle in malignant cells[44,71,72]


### 3.2. Sodium Channels

#### 3.2.1. Epithelial Sodium Channels (ENaC)

Epithelial sodium channels (ENaC) are a class of amiloride-sensitive sodium channels that are allied to sustained proliferation and invasion in a variety of cancers [73]. PcTX-1 and benzamil are amiloride analogs that act to block ENaC channel function. When targeted by these inhibitors, D-54-MG glioma cells underwent cell cycle arrest at G0/G1 and reduced the S and G2/M accumulation, suggesting that sodium influx is essential for cell cycle progression in glioma cells [64] This inward Na^+^ current found in GBM is absent in low grade gliomas (LGG) and normal astrocytes.

In gene expression analysis, sodium channels are noted to be upregulated in GBM patient samples [74]. In a study of 21 specimens, 90% showed at least one mutation in an ion channel gene, with sodium channels being associated with missense mutations. These mutations were significantly associated with shorter survival (168 days), compared to those who had no mutations (689 days). Interestingly, patients that harboured IDH1 mutations did not have any sodium channel mutations. Additionally, preferential toxicity was seen in U-87 GBM cells when targeted with the ATPase inhibitor digoxin, when compared with somatic astrocytes [75] These data should be considered with caution due to the small sample size, however they tend to support a role of sodium channels in glioma progression and the cancer cell cycle.

#### 3.2.2. Voltage Gated Sodium Channels

The voltage gated sodium channel (VGSC) α subunit family contains nine members, Nav1.1–Nav1.9, encoded by genes SCN1A–SCN11A [39], α subunits are notedly expressed in gliomas [76]. Further sequencing studies identified that SCNA8 was highly enriched in bulk tumour samples, whilst siRNA knock down of SCN8A sodium channel gene conferred reduced viability (55–62% less growth) in glioma stem-like cells (GSC) [50] conferring obvious implications in malignant progression of GSC. A study of human glioma biopsies found that higher grade gliomas were associated with expression of fewer VGSC subtypes, and lower overall expression levels. Nav1.6 is the most abundant isoform found in the CNS, however was almost completely absent in the biopsies [77].

### 3.3. Potassium Channels

K^+^ channels are a group of transmembrane proteins whose function is defined by the ability to control the selective facilitation of K^+^ efflux from the cell [78]. The resting cell has a unique pattern of ion flux, with the vast majority of ion movement being the efflux of K^+^ ions into the extracellular space. It is this sustained efflux of positive ions that creates a negative membrane potential [79]. A variety of studies have explored the role of potassium channels in neoplastic development, taking into consideration the specific subgroups of potassium channels.

#### 3.3.1. Inwardly Rectifying Potassium Channels (Kir)

Inwardly-rectifying potassium channel 4.1 (Kir4.1) expression correlates with differentiation in astrocytic cells. Characterised by a negative membrane potential of <0 mV and cell cycle exit, Kir4.1 holds a regulatory role in cell growth [80]. In Kir4.1 deficient glioma cells, generation of Kir4.1 expression lines significantly impaired growth ability via a shift in cell cycle from G2/M phase to G0/G1 quiescence. These effects could be completely reversed when inhibiting Kir4.1 channels with BaCl2 [70]. This study demonstrates that Kir4.1 is responsible for membrane hyperpolarisation sufficient to induce cell growth and maturation.

#### 3.3.2. Voltage Gated Potassium Channels

Voltage-gated potassium channels (Kv) are the largest group of ion channels; it is widely documented that Kv channels have central role in cell proliferation, by allowing progression of the cell cycle [81]. The expression of Kv channels is altered in many cancers, and their participation in neoplastic progression is well known [81]. A study of TGCA, Rembrandt and CGGA data sets identified three potassium channel genes KCNN4, KCNB1 and KCNJ10 that were found to hold a significant role in malignant progression of the tumour and were associated with overall survival in paediatric GBM (pGBM) [82]. In these samples, KCNN4 expression was upregulated, whereas the expression of KCNB1 and KCNJ10 was downregulated. Based on this genetic signature, patients were classified into high risk (three gene signature) and low risk (no signature) and findings demonstrated that the pGBM patients that were identified as ‘high risk’ of poor outcome showed an increased sensitive to chemotherapy [82]. Finally, molecular analysis of the tumours revealed that this ion channel signature harnessed a mesenchymal subtype and wild-type IDH1 preference [82]. 

The human ether-a-go-go related gene (hERG) encodes the pore-forming subunit of the K(+) channel, Kv11.1. [83]. Cancer cells typically exhibit depolarised vM, and it is speculated that hEAG have the capacity to limit these values. Depolarised vM allow large hyperpolarisations, thus driving the Ca^2+^ that is necessary for cell-cycle progression. Therefore, these channels may play a fundamental rold in cell proliferation, unlocking their oncogenic potential [84]. hERG channels have been implicated in glioma. Differential expression of hEAG1 and hERG1 is found amongst gliomas conferring to the malignant status and nature of the tumour. Kv11.1 is associated with abnormal expression in hGG [85]. Similarly, when supressed by siRNA hERG mediated apoptosis in glioma cell lines demonstrating a key role in glioma apoptosis [86].

#### 3.3.3. Calcium Activated Potassium Channels

BK channels are a group of unique voltage dependent large conductance Ca^2+^ activated K^+^ channels that function in electrical and chemical signaling [87]. BK channels in glioma cells form their own specific subclass–glioma BK channels (gBK) and are characterised by heightened sensitivity to intracellular Ca^2+^. These BK channels are upregulated in glioma biopsies, with levels of expression positively correlated with malignant grade [88]. Iberiotoxin (ibTX) is a selective pharmacological inhibitor of BK channels which has been shown to cause a dose/time dependent decrease in glioma cell number in survival assays. Further to this, inhibition of BK channels via ibTX results in S phase arrest and cellular death in human malignant glioma cells, thus demonstrating the fundamental role of gBK channels in glioma cell cycle progression [89].

### 3.4. Chloride Channels

Chloride channels are a functionally and structurally diverse group of selective channels, associated with cell volume and regulation and excitability in cardiac, neuronal, and smooth muscle cells. Due to their relationship with cell volume regulation, they are interesting targets to inhibit cancer cell motility (BR J Pharmacol, 2009). ClC-2 and ClC-3 are Cl^−^ channels that are identified to be specifically upregulated in the membranes of gliomas cells. Increased expression of these channels endow glioma cells with an enhanced route of Cl^−^ transport; in turn facilitating changes in cell shape and size during division and invasion [71].

A study utilising a gene expression array data set (accession number: GSE3289) identified 18 ion channel genes that are differentially expressed as prognostic molecular subtypes. Of the 18 channel genes identified, 16 were down regulated in HGG including the epithelial sodium channel SCN1A, anion channel VDAC, potassium channel KCNJ10 and the purinoreceptor P2RX7. However, the chloride channels CLIC1 and CLI4 were both upregulated in the high-grade cohort. A second microarray data set was employed to validate these findings (accession number: GSE4290) and the results were mirrored. Kaplan Meyer testing confirmed that tumours that harboured this 18 gene ion channel signature were associated with decreased overall survival in the cohorts compared to tumours with ion channel signature [54].

CLIC1 is found to be over expressed in GBM samples and is implicated widely in the tumorigenic capacity of GBM cells; CLIC1 silencing by shRNA reduces the proliferative and clonogenic capacities of GBM derived stem cells [56]. Further to this, clinical correlations reveal that high expression of CLIC1 is significantly associated with worse overall survival [56]. Heterogeneous mRNA expression of Cl^−^ channels genes is observed in patient glioma samples [43]. Of these genes, ANO1 a channel gene implicated in breast cancer progression, was upregulated. ANO1 is a calcium activated chloride channel that exerts its function through EGFR and CAMK pathway activation [90]. Similarly, CLIC4 an ion channel associated with poor prognosis in colorectal cancer was also upregulated. These chloride genes were found as part of an ICG signature that conferred poor prognosis in glioma [43].

Niflumic acid, a chloride ion channel blocker, inhibits glioma cell volume reduction, a process essential in the invasion of these cells. In microchannel migration assays, the cell migration index of glioma cells reduces by 43% when treated with niflumic acid. This migration was associated with a decrease in cell volume [91].

### 3.5. Calcium Channels

#### T-Type Calcium Channels

T-type Ca^2+^ channels are key regulators of the cancer cell cycle and survival. The t-type channel subunit, Cav3.2 promotes proliferation and stemness in both GBM primary cell and mouse xenografts [92]. In vivo studies of GBM murine xenografts revealed that tetralol derivatives (T-type channel blockers) can significantly slow tumour progression. Similarly, lentiviral infection of GBM cells with shRNA against Cav3.1 resulted in significant apoptosis, and in murine models, tumour size was reduced when Cav3.1 expression was silenced. TMZ resistant GBM models exhibited over-expression of Cav3.1 [93]. siRNA mediated knockdown of T-type channel subunits Cav3.1 and Cav3.2 reduced cell viability and clonogenic potential and induced apoptosis in U251 and U87 glioma cells. Further to this, cells were sensitised to ionising radiation when Cav channels were silenced [94].

### 3.6. Transient Receptor Potential Cation Channels

The expression of TRPC1 promotes cytokinesis, proliferation [95] and motility in glioma cells [96]. In human HGG, inhibition of TRPC1 via pharmacological inhibitors such as SKF96365 and MRS1845 or siRNA diminished the proliferative capacity of the cells and arrested the cell cycle. Interestingly, when stimulated with epidermal growth factor (EGF), TRPC1 relocated to the leading edge of migratory glioma cells (D54MG), suggesting a growth factor mediated role for TRPC1 in the migration of cancer cells [96]. Moreover, suppression of TRPC1 via siRNA in a lung carcinoma cell line conferred a significant decrease in cell growth associated with cell cycle arrest at G0/G1 [97]. 

TRPC1 is essential in glioma cell division, likely because of its regulatory effect on calcium signalling during cytokinesis [95], confirming it’s functional role in in the proliferation and migration of glioma cells. The entry of Ca^2+^ through TRPM channels is essential for activating Ca^2+^-sensitive K^+^ channels (KCa1.1), and initiating the machinery required for migration [11]. TRPC6 promotes glioma cell growth, clonogenicity and transition from g^2^/m phase of the cell cycle [98], proliferation and angiogenesis [99]. Similarly, suppression of TRPM7 inhibits the capacity of malignant gliomas cells to proliferate, migrate and invade [100].

Gene expression analysis by qrt-PCR found that TRPC1, TRPC6, TRPM2, TRPM3, TRPM7, TRPM8, TRPV1, and TRPV2 were significantly higher expressed in GBM patients, and that overexpression of TRP genes was positively correlated with longer overall survival [52].

## 4. Ion Channel Inhibitors as Therapeutic Targets

Ion channels present as attractive druggable targets in the treatment of a multitude of disorders. They are particularly appealing as anti-cancer agents, as molecules inhibiting the mechanism of these channels act from the extracellular space, and do not necessarily require entry into the cell. Thus, rendering one protective capacity of tumour cells–expressing drug pumping carriers–ineffective [101]. A multitude of studies have confirmed the inhibitory effect of ion channel blockers on cancer cell progression and invasion. Despite the focus of the review being specifically on high grade gliomas, evidence from studies thus far demonstrates that ion channels present as a tumour agnostic approach to most cancer therapies. However, when it comes to the utilisation of ion channel inhibitors there are grave concerns regarding the potential toxicities associated with these drugs, especially on the cardiac and nervous systems [102]. One predominant issue that arises when searching for treatments for brain cancer is over-coming the blood brain barrier [103]. The repurposing of currently available drugs is particularly appealing, Table 2 summarises some commercially available ion channel inhibitors and their therapeutic/experimental effect on patients and cell lines.

Studies looking into coupling ion channel inhibitors with nano particles [104] and utilising cavity depot delivery [105] stand at the forefront of overcoming this issue. A disease-based approach would be beneficial in this circumstance as we have a thorough understanding of the pathology of glioma. When repurposing drugs, the safety profile of the drug is already known and the pharmacokinetics in humans are well understood. The time it would take to repurpose these drugs is significantly shorter than a new drug candidate, and this comes with large cost effectiveness. Drugs targeting ion channels can exert their effect by obstructing the channel pore or binding to allosteric sites, inhibiting the channel. Similarly, amphiphilic compounds can alter the conformational state of the channel via interactions with the lipid bilayer [106].


cancers-12-03068-t002_Table 2Table 2Summary of ion channel inhibitors and their target channels repurposed for glioma cells.
**Channel**

**Tumour**

**Drug**

**References**
Cav3.2 GSC Mibefradil [92] Cav1.1, Cav1.2, Cav1.3, Cav1.4 Rat Derived GBM GBM mouse models GBM cell lines Pimozide Fluspirilene [107,108,109]Nav1.1 and Nav1.2. Human GBM Valporate Levetiracetam [110,111,112]Nav1.4 and Nav1.5 GBM cell line Riluzole [113,114]Kv1.4 GBM cell line Tamoxifen [115]Kv1.3 Human and mouse GBM biopsies Clofazimine [116,117]EAG1 Glioma Imipramine [118,119]KCa3.1 GBM cell lines Mouse GBM xenografts Clotrimazole [120,121]CLIC1 GSC GBM cell lines Metformin [122,123,124]
**Biological Toxins as Novel ion Channel Inhibitors in Cancer Treatment**

**Channel**

**Tumour**

**Toxin**

**Reference**
ClC-3 GBM, AA, Xenografts Chlorotoxin [125,126,127,128]VGSC GBM, HGG cell lines Tetrodotoxin [13,50,129]ENaC/ASIC GBM cell lines Psalmotoxin [64,66]


### 4.1. Repurposing Current Ion Channel Inhibitors

#### 4.1.1. Calcium Channel Inhibitors

T-type Ca^2+^ channels are potently blocked by neuroleptic agents (pimozide, mibefradil and penfluridol) [130]. Mibefradil (T-Type) is a drug that was initially indicated in the treatment of hypertension and chronic angina pectoris [131]. It is a calcium channel inhibitor-a benzimidazoyl-substituted tetraline that selectively binds and inhibits T-type calcium channels. It was withdrawn 10 months after the FDA approved it, as there were potential serious side effects and harmful interactions with other drugs [132]. Emerging evidence has suggested that some FDA-approved antipsychotic drugs such as penfluridol, flusirilene and pimozide (typically used in the treatment of schizophrenia) may be suitable agents to repurpose in cancer treatment. These antipsychotic compounds are potent dopamine d2 and calcium channel inhibitors that are part of the diphenylbutylpiperidine class of drugs [108].

In U87MG glioma cells, the expression of T-type calcium channel subunits (α1G and α1H) decreased during proliferation [133]. When treated with mibefradil there was a 50% decrease in channel expression, and a 700% decrease of cyclin D1 (a proliferation marker). When over expressed, the α1H subunit resulted in a 2-fold increase in cell proliferation, whereas blocking lead to 70% decrease [133]. Cav3.2 is highly expressed in human GBM specimens, significantly linking to poor prognosis (TCGA). Inhibiting Cav3.2 in GSC with mibefradil reduced growth, stemness and survival whilst also sensitising these cells to temozolomide [92]. The suppressed growth was in part due to inhibition of pro-survival pathways such as AKT/mTOR and stimulation of the BAX (pro-apoptotic) pathway. Moreover, Cav3.2 blockage increased tumour suppressor expression (TNFRSF14 and HSD17B14) and decreased the expression of the oncogenes PDGFA, PDGFB and TGFB1 [92].

Pimozide treatment of rat glioma and patient derived GBM cell lines resulted in cell cycle arrest, anti-proliferative effects and apoptosis [134]. Furthermore, pimozide may have a radio-sensitising effect, as observed in GBM mouse models, those treated with both irradiation and pimozide lived twice as long than those treated with just irradiation alone [108]. Apoptotic volume decrease is a result of fluctuations in the levels of intracellular ions, namely the loss of intracellularly K+ and is a result of decrease in ionic strength. This loss plays a critical role in the activation of the apoptotic machinery. Reduces intracellular ionic strength allows for the activation of caspases, apoptotic nucleases and the formation of apoptosomes [35].

Another neuroleptic agent, fluspirilene causes decreased cell viability, p-STAT expression and dose-dependent neurosphere formation in GSCs and GBM cell lines [109]. The efficacy of repurposing calcium channel blockers has also been demonstrated in other cancers; pimozide and penfluridol are effective in decreasing the viability in retinoblastoma and breast cancer cells [135] and disrupts cell cycle activity in pancreatic cancer cell lines [136]. Some pre-clinical evidence suggests that both pimozide and penfluridol have a synergistic effect with some mainstay chemotherapeutic agents such as cisplatin [137] and temozolomide [138].

Further, trifluoperazine (TFP) another anti-psychotic agent inhibits GBM cell growth via calmodulin type 2 (CaM2) causing the irreversible huge efflux of Ca^2+^ from the cell similarly, in xenograft models TFP caused antiproliferative effects [139]. The efficacy of TFP as an anticancer through inhibiting angiogenesis and preventing cell invasion through the chorioallantoic basement membrane [140] are well known, so there may be scope to apply this to gliomas.

Levetiracetam is an anticonvulsive drug that modulates synaptic neurotransmitter release through binding to synaptic vesicle protein SV2A. It also exerts a partial blockade of N-type calcium currents. GBM patients prescribed levetiracetam (LEV), an anticonvulsive on top of the typical treatment regime (resection, irradiation and TMZ) exhibited significantly increased overall survival (21 months in the LEV treatment and 16 months in the group without LEV). Furthermore, in MGMT methylated group, LEV treatment had a positive impact on overall survival. These data suggest that LEV treatment may have a capacity to prolong survival of patients undergoing normal GBM treatment [112]. Studying anti-epileptic drugs in glioma models may have a double pronged benefit, whilst some of these drug exhibit antitumour effects they also provide relief from seizures, a common side effect of brain tumours, thus increasing quality of life. Antiepileptics may not exert a direct effect on the tumour as anti-neoplastic drugs, however, the combination of reducing seizure related symptoms, and increasing quality of life may serve great impact in extending overall survival.

#### 4.1.2. Sodium Channel Inhibitors

Phenytoin and carbamazepine are anticonvulsant drugs that have been identified as VGSC blockers. Both have been implicated in showing efficacy in reducing proliferation and growth in melanoma, breast [141], and prostate [142] cancer cells, therefore providing good standing for applying these drugs to gliomas.

Valporate, another anti-epileptic drug that blocks Na^+^ channels, GABA transaminase, and Ca^2+^ channels that has previously reached phase III in clinical trials in the treatment of glioma. Valporate also functions as a histone deacetylase inhibitor, a class of drugs gain traction in cancer treatment [143]. Cell cycle arrest at G2/M, upregulation of proapoptotic pathways (p27, Bim, P21) and down regulation of Bcl-xL and cyclin B1 are all observed when GBM cells are treated with valproate [144]. A multi-centre metanalysis found that patients receiving treatment with valproic acid (VPA) was significantly associated with an improved overall survival (+2.4 months). However, this was largely found from older studies that focused on younger patients, so generalisation of these data need to treated with caution [110]. Evidence from clinical studies suggest that VPA is significantly linked to improved outcome in GBM patients; although the mechanism of VPA interaction with mainstay treatments is unclear, synergy is clear between VPA, TMZ and radiation. Invitro lab studies confirm clinical findings and demonstrate that VPA can induce tumour cell death, whilst preserving healthy brain tissue [111].

Riluzole is approved in the treatment of amyotrophic lateral sclerosis (ALS) and has a wide range of actions; despite this, its mechanism is poorly understood. Riluzole is thought to exert its effects by inhibiting glutamate release by inactivating voltage dependant ion channels [145]. In stem-like cultures derived from patient GBM, riluzole treatment inhibited glucose transport 3 (GLUT3), a biomarker of poor prognosis, resulting in inhibition of the HIF1 α and p-Akt pathways. Further to this, down regulation of DNA (cytosine-5-)-methyltransferase (DNMT1) was observed. DNMT1 is a gene that is responsible for hypermethylation of tumour suppressor genes and is also associated with poor prognosis in GBM patients. Similarly, the percentage of proliferating cells declined with riluzole treatment and there was a significant reduction in cell viability [113].

Riluzole attenuates TMZ induced upregulation of MGMT and enriches the anti-cancer effect of TMZ in MGMT GBM specimens. A synergistic effect of TMZ and riluzole was seen in MGMT positive cell lines, however synergy was not observed in MGMT negative lines. Significant dose and time dependant inhibition of GBM cell growth was clear [114].

#### 4.1.3. Potassium Channel Inhibitors

Tamoxifen is a nonsteroidal mixed antiestrogenic agent that acts as an oestrogen receptor antagonist [146]. Tamoxifen has wide-spread indications in the hormonal treatment of breast cancer and acts also as a multichannel blocker that inhibits the conductance of several potassium channels [147]. Tamoxifen is already extensively used as an anti-tumour drug, that demonstrates no significantly toxic side effects, and importantly can cross the blood brain barrier [148]. In many cell types [149,150], tamoxifen has been shown to inhibit both volume-activated Cl^−^ currents and various ligand and voltage-gated cation channels [151]. Similarly, tamoxifen blocks K^+^ channel mediates neuroblastoma proliferation and has inhibitory effects on delayed rectifier K^+^ currents [151]. Recently, work on repurposing tamoxifen has indicated that it may also exert a chemotherapeutic effect on high grade glioma; in cell lines, tamoxifen was cytotoxic, inducing apoptosis. More specifically, this study was carried out on the TMZ resistant cell lines U251 and BT325, presenting an alternate therapy for those resistant to mainstay chemotherapy drugs [152]. Tamoxifen induced apoptosis and exerted cytotoxic effects in rat glioma cell, in a concentration and dose dependant manner [115]. Due to its inhibitory effects on PIP2 sensitive channels, it is suggested that that tamoxifen inhibits the Kv7.2/Kv7.3 by obstructing PIP2-channel interaction, however the exact mechanism by which tamoxifen inhibits K+ channels in unknown [153].

Kv1.3 voltage gated potassium channel is expressed in the mitochondria of both mouse and human models; treatment with novel Kv1.3 inhibitors PAPTP and PCARBTP both induce cell death in glioma cell lines [117]. Similarly, clofazimine an antimycobacterial indicated in the treatment for leprosy, exhibited inhibitory effects [117]. Clofazimine has been found to block Kv1.3 action by two mechanisms; both of which contribute to reduced capacity or inactivity of the channel: (i) a use-dependant block for open channels during long periods of depolarisation, this results in an accelerated inactivation of K^+^ current. Mechanism (ii) blocks closed, deactivated channels after said channels were opens briefly [116].

In vitro and in vivo studies confirmed the promise of imipramine in the treatment of glioma. Imipramine is a tricyclic antidepressant which is used mainly in the treatment of depression, and actively inhibits voltage gated potassium channels. In vivo, a combination of imipramine and doxorubicin conferred an anti-invasive effect [118], whereas imipramine in combination with ticlopidine suppressed autophagy signaling pathways, and resulted in cell death [119].

Clotrimazole, the anti-fungal agent selectively inhibits calcium sensitive potassium channels, in particular KCa3.1 [154]. In U87MG glioma cell lines, clotrimazole induces apoptosis, sensitises tumour cells to radiation, and arrest cells in late G1 phase. Due to the nature of these results, there is potential in the use of clotrimazole as a radio-sensitising agent in GBM [120]. Mouse glioma xenografts treated with clotrimazole underwent significant inhibition of tumour growth, and when used in combination with cisplatin chemotherapy, prolonged overall survival was observed [121].

#### 4.1.4. Chlorine Channel Inhibitors

Evidence from various epidemiological and preclinical studies suggest that biguanides possess anti-neoplastic properties. Biguanides are a class of drugs that have a wide range of medical implications i.e., antimalarials, with functional group of two guanidines linked to a nitrogen. Metformin, a type 2 diabetic agent [155] is the most promising biguanide to reposition as an anti-cancer agent.

Metformin acts synergistically with TMZ to inhibit proliferation and expansion of GSCs in culture [122] and reduces cells’ acquired resistance to TMZ [156]—an effect often seen in GBM patients contributing to poor prognosis. Treatment with metformin significantly reduces sphere forming units in 3D models of both GBM (U251) and neuroblastoma (SH-SY5Y) cell lines by targeting GSCs in these populations [124]. Similarly, further studies show reduced survival, proliferative capacity and synergism both in vivo and in vitro [123,157,158].

Metformin promotes differentiation of glioma initiating cells into non-tumourigenic cells via FOX03 activation. Further to this, FOX03 activation was initiated via AMP-activated kinase activation. Metformin treatment depleted the self-renewing capacity of tumour cell population and inhibited tumour formation, suggesting a viable therapeutic strategy to inhibit glioma cells via AMPK-FOX03 with metformin [155].

##### Clinical Trials

A recently completed phase I drug expansion trial aimed to determine the maximum tolerated dose of mibefradil when used in combination with irradiation. Patients received mibefradil, dose escalated from 150 mg/day and radiation consisted of 5 fractions of 600 cGy each, over a two-week period, followed by re-resection surgery. One patient had a complete radiographic response and interestingly, in 2 participates, mibefradil was detected at micromolar levels in GBM tumour tissue. The study demonstrated that pharmacologically effective concentrations of the drug are achieved in resected brain tumour tissue [159].

A phase 2 trial of valporate (VPA) in addition to the combination of TMZ + RT in patients with newly diagnosed GBM demonstrated an extension of the median survival from 14.2 to 29.6 months, with a very low side effect profile [160].

Similarly, a multi-institutional Phase II trial of A Phase II of Inhibitor Valproic Acid in Combination with Temodar and Radiation Therapy in Patients with High Grade Gliomas is currently on going.

The association of metformin use and survival was assessed in a pooled analysis of patient data from 1731 individuals from the randomized AVAglio, CENTRIC and CORE trials. Metformin in combination or as a stand-alone therapy was not found to be significantly associated with OS or PFS. However, it was noted that additional studies with specific tumour characteristics may be of benefit to target metabolic vulnerabilities [161].

A randomized interventional phase II clinical trial for the efficacy and safety of low dose temozolomide plus metformin as combination chemotherapy compared with low dose temozolomide plus placebo in patient with recurrent or refractory glioblastoma is currently recruiting (NCT03243851).

#### 4.1.5. Issues with Targeting Ion Channels

Despite the clear success of ion channel modulators in a wide range of pathophysiological settings, these drugs are yet to be employed routinely in the treatment of cancers. Mounting studies and considerable in vitro evidence suggests that they may be efficacious in a combinational approach with mainstay cancer therapies, however there is a great need for further in vitro and mechanistic analysis [39,102].

Ion channels are ubiquitously expressed in a plethora of cell and tissue types, so a major concern comes from the potential systemic effects of using these inhibitors [102]. For example, cardiac cells express high levels of ion channels of all classes, and many ion channels targeting drugs have been withdrawn from clinical trials due to cardiac toxicities. Ion channel splice variants and subunits and the employment of local delivery systems may be the key to developing specific tumour inhibitors [39].

## 5. Electrotherapy

The scarcity of current effective therapies for HGG has led to new treatment options being the focal point of much research. Electrotherapy is one such treatment option that is being widely explored, with use currently approved across Europe and the U.S. The Optune™ system developed by Novocure Ltd. (Haifa, Israel) is a novel FDA-approved electrotherapeutic treatment for primary and recurrent adult GBM. The Optune™ device is indicated for patients 22 years and older that have a histologically confirmed case of supratentorial GBM (WHO Grade IV astrocytoma) [162]. The use of Optune™ as a therapy is approved in combination for patients who have received maximal surgical resection and with those who have received concomitant TMZ and radiotherapy [163]. The Optune™ device works by generating alternating electric fields delivered directly to the patient–coined tumour treating fields (TTF).

The field generator delivers alternating electric fields at a recommended 200 kHz [164] through the insulated transducer arrays (attached to the patient’s shaved scalp) with a minimum field intensity of 1.0 V/cm [165]. These tumour fields are delivered throughout the tumour in a non-invasive manner. The optimal array placement is calculated by NovoTAL™ (Novocure, Ltd.), a purpose-made computational tool, which uses simulation software to optimise the field intensity within the tumour, accounting for variables such as head size and shape, resection cavity and swelling. Thus, delivering maximum therapeutic level relative to tumour burden [166]

TTF are a non-invasive antimitotic therapy, that in the EF-14 phase III clinical trial, showed significant improvement in both overall and progression free survival when used in combination with maintenance TMZ when compared with TMZ alone [163]. Similarly, the NovoTTF vs. ‘physician’s choice: chemotherapy’ trial revealed that although there was no significant improvement in the use of TTF in the absence of chemotherapy approach, TTF did produce comparable efficacy and activity to standard chemotherapy regimens, with toxicity and quality of life clearly favouring TTF [167]. Interestingly, post hoc analysis of the EF-14 trial found that TTF plus TMZ was associated with an increase in both progression free survival and overall survival, regardless of MGMT promotor methylation status [168]. Although there is no confirmed direct mechanism by which TTF and TMZ act in synergy, we can speculate that this may be a result of synthetic lethality, due to TTF interfering with the DNA repair process directly to reduce the amount of MGMT directed DNA repair. Alternatively, TTF may act as a trigger in forcing TMZ induced senescence in GBM tumour cells.

Better clinical outcome was associated with compliance of patients (the average monthly use of the device) [163]. A review by Branter et al. reveals that there are clear indications for TTF as a combinatorial therapy with both mainstay and novel therapeutics, however there is limited efficacy when used as a monotherapy. Despite there being some descriptive mechanistic preclinical evidence there is a significant lack of study into the mechanism of TTF in complex models [162].

It is understood that TTF show efficacy by disrupting mitotic capacity of the cell by inhibiting spindle apparatus formation [169]. However, there are clear morphological changes and this proposed mechanism does not fully explain or account for the genetic changes observed. The effect of TTF on ion channels may hold the key to understanding the mechanism behind its anti-neoplastic effects, a review by Zhu and Zhu explains the electrophysiological principles behind TTF action [170]. One of the proposed mechanisms of Optune is via the antimitotic effects of TTF. Via the generation of dielectrophoretic forces and the subsequent the disruption of dipole alignment during cytokinesis; TTF preferentially inhibit cancer cell proliferation. This occurs as a direct result of the interference of microtubule polymerisation, and their further assembly with other polar molecules during mitosis. Polar molecules are susceptible to electrical manipulation, and it is thought that TTF interact and exert their effect on these polar molecules during mitosis [171,172]. At the start of metaphase pairs of centromeres are captured by microtubules, orienting them towards their specific poles. Sister chromatid separation (via cytokinesis) [173] is a direct result of securin and cyclin B mediated degradation by Cdc20 and APC [174]. This formation of this destruction complex if wholly dependent on correct localisation and function of microtubules at both anaphase and metaphase [171,173]. Errors in this intricate process, particularly errors following anaphase are irrevocable. Cancer cells depend on mitotic competency and when this is compromised i.e., by errors committed in anaphase due TTF exertion, a magnitude of cell fates i.e., aberrant mitotic exit [175], apoptosis and mitotic catastrophe can occur [171,176].

A key process that TTF targets is tubulin polymerisation a process by which microtubules undergo constant cycles of polymerisation and depolymerisation. Tubulins are amongst some of the most polar molecules and are highly susceptible to disruption by TTF, promoting consistent depolymerisation. TTF force polar molecules to align with the electric field, causing misalignment of the individual tubulin subunits resulting in microtubule disruption [171,177].

## 6. Conclusions

In conclusion, membrane potential is a crucial biophysical signal that modulates cellular functions such as proliferation and differentiation even in non-excitable cells. Therefore, the plethora of cellular ion channels that are expressed must be tightly modulated by a finely tuned system to enable homeostatic maintenance of V_m_. It has been well established that cancer cells exhibit distinctive properties in terms of their bioelectrical capacity, notably, cancer cells harness a depolarised V_m_ which promotes a proliferative phenotype. Furthermore, recent studies are confirming the functional role of V_m_ in the metastatic cascade of cancer cell invasion and migration, conferring prognostic value. Membrane potential could soon be considered a clinical marker for both cancer detection and prognosis. Glioma cells have been shown to exhibit unique ion channel gene expression that aids in the proliferative capacity of these cells. The prognosis of GBM is particularly abysmal due to the cells capacity to invade and migrate into the surrounding brain. Ion channels are heavily implicated in altering the morphology of cancer cells, such as the spindle shape of glioma cells that aids in migrating through small extracellular spaces. It is clear that ion channels are intrinsic in the malignant capacity of gliomas, thus representing a potential biomarker and treatment target for GBM. There is clear preclinical efficacy in the multimodal use of electrotherapy pertaining to GBM however the mechanism by which it kills cancer cells is unclear, leaving a large platform to assess this and the role that ion channels may play in potentiating electrotherapy. Further research is needed in this field; ion channels present themselves as interesting candidates as the mechanistic key to Optune therapy, thus should be exploited to further potentiate this treatment.

Due to the lack of effective, life prolonging treatment for HGG, there is clear rationale in pursing the role of ion channels in the progression of glial cancers and consequently transforming them into therapeutic targets for cancers. Ion channel inhibitors already exist in a multitude of FDA approved drugs, thus exploring the repurposing of these pharmacological inhibitors for cancer treatment is a promising direction for future research.

## Figures and Tables

**Figure 1 cancers-12-03068-f001:**
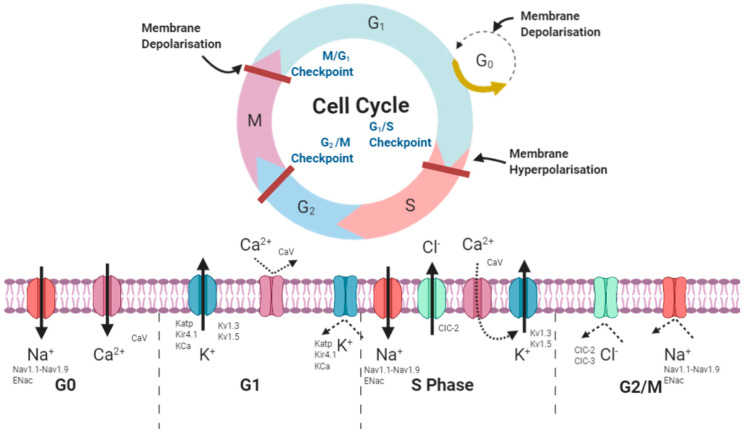
Schematic representing the activities of ions during phases of the cell cycle. G0 phase is associated with opening of Na+ and some Ca^2+^ channels, causing and inward positive influx, deploarising the cell. As the cell moves into G1^–^ cell growth phase, K^+^ channels open and positive ions move out of the cell into the extracellular space. Ca^2+^ channels close preventing inward positive flux. As the cell begins to repolarize and moves towards the DNA replication phase, K^+^ channels close preventing outward positive flux, this event is necessary to promote G0/G1 to S phase transition. During S phase, Na^+^ channels open again alongside Cl^−^ channels. Finally, as the cell transitions to G2/M phase, both Cl^−^ and Na+ channels close preventing any further ion influx or efflux. Created with BioRender.com.

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
