# Peer review of "Ion Channels as Therapeutic Targets in High Grade Gliomas"

_cancers, 2020, doi:10.3390/cancers12103068_

Round 1
Reviewer 1 Report
The manuscript entitled “Ion channels as therapeutic targets in high grade gliomas” by Griffin M. et al. carefully reviewed the role of the ion channels in glioma development. The review introduced connection between the ion channels and cell cycle, glioma migration and targetable potentials of ion channels. It emphasized the types of ion channels in glioma and their inhibitors. Overall this is a well written review article. The premise of the work is very interesting and should offer insights into the essential roles of ion channels, however in its present version, the manuscript requires several significant areas of improvement before consideration for publication.
1- Molecular regulation (genetic and epigenetic) of ion channels should be mentioned in the review.
2- The role of ion channels in the cell cycle was emphasized in the beginning of manuscript, however it was not followed up properly. Connections between specific type of ion channels and glioma progression should be made.
3- Besides the repurposed inhibitors, clinical trials of ion channel inhibitors (if any) should be summarized.
4- Showing of regulation of ion channels in glioma as a figure will be better, and it can be emphasized to targetable pathways with inhibitors.
6) It would be more interesting if authors can provide one separate Table or Figure showing role of ion channels in gliomas. The manuscript will definitely attract more readers if more figures are included.
8) The figure and their accompanying legend is very blurry. I would recommend presenting this in a different way to make it more clear for the reader. Please check final resolution to facilitate ease of understanding and reading.
9) There are some points either discussed haphazardly or overlooked, need to be discussed properly. For example, detailed information about electrotherapy is needed.
10) Table 1. Summary of ion channel inhibitors and their target channels repurposed for glioma cells, for this table, a figure can be designed.

Author Response
| Molecular regulation (genetic and epigenetic) of ion channels should be mentioned in the review. | Identified some transcriptional mechanisms line 85-98. The authors have searched the literature and not enough data specific to GBM found. One of the aims of this review is to promote further research and identify gaps in literature, and epigenetic regulation of ion channels is one such important area. | |
| The role of ion channels in the cell cycle was emphasized in the beginning of manuscript, however it was not followed up properly. Connections between specific type of ion channels and glioma progression should be made. | Specific glioma ion channels have been added to Figure 1 and edits have been made throughout the paper refering back to the cell cycle where applicable. Table 1 has been created showing ion channels specific to glioma and the role they play i.e. cell cycle progression. | |
| Besides the repurposed inhibitors, clinical trials of ion channel inhibitors (if any) should be summarized. | Line 477 Physicians choice clinical trail for optune. Line 535 - Clinical trial data for ion channels added. | |
| Showing of regulation of ion channels in glioma as a figure will be better, and it can be emphasized to targetable pathways with inhibitors. | We have searched literature and found little to produce sufficient figure, this is one such area that the authors are interested in exploring further. | |
| It would be more interesting if authors can provide one separate Table or Figure showing role of ion channels in gliomas. The manuscript will definitely attract more readers if more figures are included. | Table 1 - specific ion channels and their role in glioma/ experimental findings made | |
| The figure and their accompanying legend is very blurry. I would recommend presenting this in a different way to make it more clear for the reader. Please check final resolution to facilitate ease of understanding and reading. | we have ensured all images are atleast 300DPI and 8x5cm | |
| There are some points either discussed haphazardly or overlooked, need to be discussed properly. For example, detailed information about electrotherapy is needed. | Paper reviewed and care was taken to ensure all points made were made in full with adequte details. Detail added to electroherapy explaining mechanism and some clinical data- 525 onwards | |
| Table 1. Summary of ion channel inhibitors and their target channels repurposed for glioma cells, for this table, a figure can be designed. | We thank the reviwer for the suggestion. This table was kept as a figure could not be conceptualised |
Reviewer 2 Report
This is an interesting and topical review, summarizing our knowledge on ion channels in glioblastoma therapy. Treatment of GBM, which rests on resection, TMZ and radiation, is still unsuccessful and alternative or supporting therapies are needed. Treatments targeting ion channels seem to be promising.
Points to be considered during review:
- hERG channels were briefly addressed. In Table 1, EAG1 = Kv10.1 or KCNH1 are mentioned, also the inhibitor imipramin. However, it is not explained in the text. Please describe the family of hERG also, including KCNH2 or Kv 11.1) and its role in GBM responses (see https://www.tandfonline.com/doi/full/10.1080/09687688.2020.1729428).
- line 114: ion exchange processes have impact on apoptosis; line 322: Pimozide causes radiosensitization and line 331: sensitization to temozolomide. There is no molecular-biological explanation, which is unsatisfactory. Please provide at least a hypothetical model on how ion channels are linked to the known apoptosis pathways.
- line 358: Interestingly, valproic acid was shown to impact TMZ-induced cytotoxicity in malignant melanoma cells through downregulation of RAD51/FANCD2 mediated homologous recombination DNA repair pathway (Krumm et al., Cancer Res., 2016; DOI: 10.1158/0008-5472.CAN-15-2680). I'm wondering whether similar data are available for GBM cells.
- line 379: what means "inhibition of GBM cells" ?
- line 389: "... TMZ resistant cell lines A172 and others...". A172 is MGMT lacking and strogly responding to TMZ, compared to U87MG. I wouldn't consider A172 a resistant cell line. There are numerous papers using A172 for apoptosis and cell death studies.
- 382-391: In this paragraph, tamoxifen is described as estrogen receptor antagonist, but not a word is said about its possible effect on potassium channels. Is tamoxifen a channel inhibitor, and if so, why?
- line 447: Optune is effective in patients who have received TMZ/radiotherapy. Is there a difference between MGMT promoter methylated versus unmethylated patients?
- line 452: TTF is in combination with TMZ effective. Why? Please add some words, even if it is speculative. Does TTF force the TMZ-induced cell death pathway or act it as a trigger of senescence?
Author Response
| 1. hERG channels were briefly addressed. In Table 1, EAG1 = Kv10.1 or KCNH1 are mentioned, also the inhibitor imipramin. However, it is not explained in the text. Please describe the family of hERG also, including KCNH2 or Kv 11.1) and its role in GBM responses (see https://www.tandfonline.com/doi/full/10.1080/09687688.2020.1729428). | Line 234 - hERG channels addressed with reference to KV11.1 in glioma. | |
| 1. line 114: ion exchange processes have impact on apoptosis; line 322: Pimozide causes radiosensitization and line 331: sensitization to temozolomide. There is no molecular-biological explanation, which is unsatisfactory. Please provide at least a hypothetical model on how ion channels are linked to the known apoptosis pathways. | Added to line 361 - AVD in relation to K+ ions. Added to line 129 - example of calcium channels and apoptosis | |
| line 358: Interestingly, valproic acid was shown to impact TMZ-induced cytotoxicity in malignant melanoma cells through downregulation of RAD51/FANCD2 mediated homologous recombination DNA repair pathway (Krumm et al., Cancer Res., 2016; DOI: 10.1158/0008-5472.CAN-15-2680). I'm wondering whether similar data are available for GBM cells | Interesting study and has sparked th authors interests. At this current time, we cannot find data r.e. GBM cells and RAD51/FANCD2 pathways with specific reference to TMZ + VPA Valporate sensitises cells to TMZ | |
| line 379: what means "inhibition of GBM cells" ? | Now line 401 - mistake corrected to "GBM cell growth" | |
| line 389: "... TMZ resistant cell lines A172 and others...". A172 is MGMT lacking and strogly responding to TMZ, compared to U87MG. I wouldn't consider A172 a resistant cell line. There are numerous papers using A172 for apoptosis and cell death studies. | Mistake corrected | |
| 382-391: In this paragraph, tamoxifen is described as estrogen receptor antagonist, but not a word is said about its possible effect on potassium channels. Is tamoxifen a channel inhibitor, and if so, why? | Additions to line 427 and 436 describing suggested mechanism. | |
| line 447: Optune is effective in patients who have received TMZ/radiotherapy. Is there a difference between MGMT promoter methylated versus unmethylated patients? | Addressed on lines 540 - There was no difference found between MGMT +/- groups in EF-14 trial, but could not find MGMT +/- specific trial. | |
| line 452: TTF is in combination with TMZ effective. Why? Please add some words, even if it is speculative. Does TTF force the TMZ-induced cell death pathway or act it as a trigger of senescence? |
I am not aware of any direct evidence for any of these mechanisms. However, speculative notes have been added r.e. MGMT regulated DNA repair. Line 541. This is an area of interest to the authours. We would like to thank you for all of the above constructive comments. |
Reviewer 3 Report
This review summarized the current evidence of ion channels in the gliomagenesis and the potential therapy using channel inhibitors for HGG. I have the following suggestion for the authors:
1) Change the "Introduction" to "Glioma".
2) Please add a figure to summary the channels and related glioma types. This will help the readers to understand this review.
3) Please add a paragraph to summarize the clinical advance/application or clinical trial of the therapies mentioned in this review.
4) Please revise the manuscript to correct the typo and make the manuscript easier reading. Such as "5. Electrotherapy (TTF)" to "5. Electrotherapy".
Author Response
| 1) Change the "Introduction" to "Glioma". | "Introduction" changed to "glioma" line 30 | |
| 2) Please add a figure to summary the channels and related glioma types. This will help the readers to understand this review. | Table of specific ion channels and their role in glioma/ experimental findings made | |
| 3) Please add a paragraph to summarize the clinical advance/application or clinical trial of the therapies mentioned in this review. | Physicians choice clinical trail and ef14 trials expanded for optune. Line 429 - Clinical trial data added for ion channels. | |
| 4) Please revise the manuscript to correct the typo and make the manuscript easier reading. Such as "5. Electrotherapy (TTF)" to "5. Electrotherapy". |
"Electrotherapy (TTF) changed to "Electrotherapy"
The authours would like to thank you for your constructive comments |